# Optimizing the Use of Cultivated Land in China’s Main Grain-Producing Areas from the Dual Perspective of Ecological Security and Leading-Function Zoning

**DOI:** 10.3390/ijerph192013630

**Published:** 2022-10-20

**Authors:** Chengxiu Li, Xiuli Wang, Zhengxin Ji, Ling Li, Xiaoke Guan

**Affiliations:** 1College of Resources and Environment, Henan Agricultural University, Zhengzhou 450002, China; 2Henan Engineering Research Center of Land Consolidation and Ecological Restoration, Zhengzhou 450002, China; 3College of Land Science and Technology, China Agricultural University, Beijing 100193, China; 4Social Development Research Center, Zhengzhou University of Light Industry, Zhengzhou 450002, China

**Keywords:** ecological security, multifunctional assessment, cultivated land protection, differentiated control measures

## Abstract

In order to achieve the coordinated development of ecological protection and cultivated land use, ecological security and cultivated land use functions (CLUFs) in the study area were evaluated by constructing a comprehensive evaluation index system. The leading CLUFs were measured, and it was determined to use the normalized revealed comparative advantage (NRCA) index. The spatial superposition analysis of the ecological security level and the leading CLUFs was carried out to obtain the zoning of the coordinated development of ecological security and cultivated land use, and differentiated utilization strategies were proposed for different zones. The results of this study showed the following: (1) The ecological security level of cultivated land in Yuanyang County can be divided into high, medium, and low security levels, accounting for 30.68%, 43.42%, and 25.9% of the total cultivated land area, respectively. The overall ecological security level is high. (2) The current cultivated land use mainly has a production function, accounting for 38.39% of the total cultivated land area, the leading CLUFs that are 34.16% of the cultivated land are the ecological function, and 27.45% of the cultivated land is the living function. (3) The spatial superposition analysis of the ecological security level and leading CLUFs yielded four zones of cultivated land use enhancement, including a production core zone, and different control strategies were proposed for the different zones. These strategies may help to fully realize the multifunctionality of the cultivated land without compromising ecological protection. Implementing differentiated protection for cultivated land in different use zones can achieve the coordinated development of ecological protection and cultivated land use, thus promoting the sustainable use of cultivated land resources.

## 1. Introduction

As the most important means of production in human survival and development, cultivated land plays a very important role in guaranteeing people’s livelihoods, maintaining social stability, and maintaining ecological balance [1]. Since the 20th century, rapid economic development has led to tremendous social and environmental changes. Phenomena such as global population growth, economic growth, and higher consumption levels of agricultural products have directly exacerbated the contradictions between cultivated land conservation and social development [2,3]. Some developing countries are facing the pressure of rapid industrialization and urbanization, which require more cultivated land, while they also face the problems of population pressure, pollution of agricultural production, and ecological damage of cultivated land. As the world’s most populous country, China needs to feed 22% of the world’s population, but it only contains 7% of the world’s cultivated land [4,5], which makes food production a rough challenge. Moreover, whether food security can be effectively ensured not only affects the livelihood of the general public, but it also has bearing on the global food pattern. Driven by population pressure, food pressure, scarcity of available land, and the increasing demand for agricultural land, the high consumption of fossil energy sources, the high pollution created by pesticides and fertilizers, and the high emissions of agricultural greenhouse gases from cultivated land use have led to the gradual emergence of environmental damage and the unbalanced development of the cultivated land use to obtain higher food production and economic benefits [6]. Although this has improved agricultural productivity and labor efficiency, it runs counter to the concept of sustainable and ecological agricultural development, leading to constant conflicts between cultivated land use and ecological protection and affecting the coordinated development of both. In fact, the Chinese government has been continuously concerned about the protection of cultivated land. As early as 2006, a policy was introduced to ensure the basic stability of the quantity of cultivated land and to improve its quality; meanwhile, a series of systems were formulated to protect cultivated land, such as the compensation system for cultivated land occupation, the dynamic balance system of the total cultivated land, and the basic cultivated land protection system [7,8]. However, these systems were only aimed at protecting the quantity and quality of cultivated land. Regarding the ecological protection of cultivated land, the only effort was to establish an ecological compensation system for cultivated land or to enhance the ecological service functions of cultivated land according to local conditions to promote sustainable cultivated land use resources. Furthermore, there are no proven measures to ensure the coordinated development of the ecology and cultivated land use [9]. Since 2019, the new crown epidemic and extreme climate disasters have posed great challenges and impacts on China’s grain production. The injection of ecological security into the process of cultivated land use is consistent with current research trends and social development needs. Therefore, how to rationally optimize the allocation of cultivated land resources on the premise of ensuring ecological security is a problem that needs to be solved at this stage.

Previous studies provide good references for the coordinated development of ecological protection and cultivated land use, but much effort is still needed to develop a suitable method for the comprehensive evaluation of both. Although the PSR [10,11] model and the DPSIR [12,13] model are quite mature among the current methods of researching the evaluation of the ecological security of cultivated land, their shortcoming lies in the selection of the indicators. The restricted evaluation indicators of the PSR model are mostly statistical indicators related to the economy, environment, and nature [14,15]. Although they can respond to the relationship between the cultivated land ecosystem and human social development to a certain extent, they ignore the spatial pattern of the cultivated land itself and the influence of the surrounding environment [12,16]. In terms of cultivated land utilization, in recent years, with the proposal of the concept of “production-living-ecological (PLE)” space, the core of land use optimization has gradually changed from “quantity-space” to “structure-function”. Therefore, how to maximize the utilization CLUFs is the key to the optimal allocation of cultivated land resources. At the present stage, the calculation and evaluation methods of CLUFs mainly include the index system method, physical measurement method, monetary measurement method, and energy analysis method [17]. The physical measurement method evaluates the CLUF system from the perspective of physical quality. It is difficult to achieve a comprehensive evaluation of different CLUFs due to the non-uniformity of the scale; the monetary measurement method quantitatively evaluates the CLUFs from the perspective of the monetary value, with the calculated ecological and social functions often exceeding the actual value. The energy analysis method converts the CLUFs into standard solar values to compare and evaluate the size of each CLUF. It is a complicated calculation process and makes it difficult to achieve the spatial quantification of multifunction cultivated land; thus, the index system method is the most commonly used [18,19,20].

The purpose of this study was to reconcile, weaken, and resolve the conflict between ecological conservation and cultivated land use. Compared with previous studies, we analyzed the essential properties of the cultivated land from the perspective of ecological security and evaluated the innate formation elements, late constitutive elements, and external threats of the cultivated land from the foundation-support-threat perspective. Additionally, in consideration of the multifunctionality of cultivated land as a composite function, we adopted the normalized revealed comparative advantage (NRCA) index proposed by Yu et al. [21,22,23]. In this way, we were able to measure the leading CLUFs to propose corresponding recommendations and countermeasures for cultivated land use. Finally, in this study, the results of both the ecological security and CLUF evaluation were superimposed. Without compromising ecological security, function-oriented zoning was suggested for cultivated land use, which improves the efficiency of resource use more effectively than the blind all-around protection approach. It not only highlights the food production function of cultivated land compared to monolithic land use, but it also takes into account food production security and the landscape ecological function. In addition to this, we provided each zone with the necessary resources to meet its needs. Finally, we provided theoretical guidance for each zone and proposed feasible suggestions with local characteristics to break the spatial conflicts among production, ecology, and life and to guide the development of regional cultivated land in the direction of balance between ecological protection and economic benefits.

## 2. Materials and Methods

### 2.1. Overview of the Study Area

Yuanyang County is located in Xinxiang City, Henan Province (Figure 1). There are 16 townships, 3 offices, and 583 administrative villages under its jurisdiction. Yuanyang County has a warm temperate continental monsoon climate and is located on the Yellow River Alluvial Plain in northern Henan. Its unique geographical growth conditions make it a national commercial grain production base and a demonstration area for high-quality rice cultivation. Yuanyang County has a special geographical location. Located along the Yellow River, Yuanyang is the county with the largest wetland area of the Yellow River in Henan Province. Its forest coverage is the first in the plain area of Henan Province, and its ecological status is prominent. Additionally, the cultivated land area in Yuanyang County accounts for 61.96% of the total land area, which is more than 10% higher than the provincial average, and it is all located at a slope below 2 degrees. Yuanyang County has adopted measures such as introducing rice from the Yellow River, pressing alkali and sand, and improving soil nutrients. Most plots are planted with two crops of rice and wheat. The planting area of high-quality rice exceeds 300 km^2^. The rice produced was the first grain in Henan Province to obtain green food certification and earn foreign exchange through export. As it is located along the Yellow River, the effective irrigation area is 499.7 km^2^, the length of the irrigation canal is 2084.36 km, and the farmland infrastructure is perfect, making it a high-yield and efficient agricultural area for producing high-quality wheat and rice. As a major grain-producing county in Henan Province and even in the whole country, Yuanyang County has the important tasks of both ecological protection and grain production. The Chinese government formulated a major national strategy for the ecological protection and high-quality development of the Yellow River Basin in 2019, proposing to strengthen the ecological protection and management of the basin and to promote the high-quality development of the entire basin. Therefore, Yuanyang County was used as a study area to promote the coordinated development of ecological protection and cultivated land use by rationalizing the cultivated land use from the perspective of ecological security, which is of great significance for the ecological protection and high-quality development of the entire basin.

### 2.2. Research Methods

#### 2.2.1. Research Frame

The proposed framework formulates policy recommendations for the ecological protection and utilization optimization of cultivated land by evaluating ecological security and the leading CLUFs (Figure 2). This framework was divided into four steps. Firstly, a comprehensive evaluation of the ecological security level of cultivated land was conducted. We analyzed the essential properties of the cultivated land from the perspective of ecological security and evaluated the innate formation elements, late constitutive elements, and external threats of the cultivated land from the foundation-support-threat perspective to evaluate the level of ecological security of the cultivated land. Secondly, from the perspectives of production, living, and ecological, we evaluated the CLUFs and used the NRCA model to calculate the leading CLUFs. Thirdly, the results of both the ecological security and CLUF evaluation were superimposed. We used the NRCA model again to measure the leading zoning of each township and adjusted the cultivated land use zoning by using the nearest neighbor method and the maximum patch principle with the leading zoning of each township as the limiting factor. Finally, we proposed differentiated utilization measures and policy recommendations for different regions.

#### 2.2.2. Ecological Security Evaluation

Based on the foundation-support-threat perspective, we selected the three indicators of natural background, ecological support, and ecological risk to construct an evaluation system for the ecological security of the cultivated land (Table 1).

Considering that the topography of Yuanyang County is flat, and there are no significant spatial variations in the rainfall and temperature within the county, large scale indicators such as rainfall and temperature were excluded when selecting indicators. Ecological environment characteristics such as the soil organic matter content, profile pattern, groundwater level, and pH value were selected as evaluation indicators [24]. A high soil organic matter content can significantly improve the physical properties of the soil and enhance the fertility and buffering properties of the soil. The soil profile pattern reflects the permeability, leaching rate, and uniformity of the material migration in the soil profile, which has an important impact on the accumulation of soil organic matter, water and fertilizer retention, salinity hazards, and water drainage. The groundwater level reflects more than just the ease of groundwater utilization by crops. In the low-lying areas of the plain, the groundwater level also has a significant influence on soil salinization and waterlogging. While a high groundwater level is convenient for agricultural water, it may lead to the problem of salt returning to the surface. The soil pH reflects the ability of the soil to buffer the acidity and alkalinity, and the suitable pH range differs from region to region and from crop to crop. A suitable pH value can create a more stable living environment for crop growth and soil organism activities.

For ecological support, we selected vegetation cover, vegetation net primary productivity, and leaf area index as indicators [25]. Vegetation cover refers to the proportion of the vertical projection of the canopy and branches of all of the vegetation (trees, shrubs, grasses, and crops) in the growing area, and it can describe the quality and stability of the regional ecological environment well. The net primary productivity (NPP) of vegetation reflects the productivity of the plant communities under natural environmental conditions, and it can be used to evaluate the coordination of the ecosystem structure and function. The leaf area index is a good indicator of the growth of crop communities and the stability of ecosystems, and it is a measure of the vitality of vegetation communities and their environmental effects.

For ecological risks, we selected the soil erodibility index, groundwater utilization, and built-up land expansion index [26]. The soil erodibility index indicates the ease of separation and movement of the soil material in a soil body under the action of external forces. The higher the value of the soil erodibility index, the more susceptible the soil is to erosion. Excessive exploitation of groundwater not only causes a series of ecological problems such as the reduction of biological communities and the decay of spring flow due to the continuous decline of the groundwater level, but it also contributes to the occurrence of geological disasters such as ground subsidence and cave-ins. The ecological and environmental problems brought about by the expansion of built-up land during the urbanization process are inevitable, including the risk of heavy metal pollution caused by the discharge of waste gas, wastewater, and solid waste during industrial production, as well as pesticides, fertilizers, mulch, livestock breeding manure, and other agricultural surface pollution sources brought about by the large demand of new urban area for surrounding agricultural and livestock products. In addition, the hardening of the surface caused by human engineering activities leads to many ecological problems, including increased difficulty of the infiltration of surface water, a lack of groundwater recharge, drastic changes in the surface temperature, a poor buffering effect, and the destruction of the microbial environment, which seriously affect the ecological stability of the area.

Ecological security levels were classified with reference to the relevant literature [27,28,29].

#### 2.2.3. CLUFs Evaluation

##### Evaluation of Individual CLUF

The agricultural production function is the most basic CLUF and is an essential attribute of cultivated land [30], while the yield is the most direct manifestation of the production function of cultivated land. A road network between fields can not only improve the production efficiency of cultivated land, but it can also facilitate the construction of ecological farmland. Therefore, we measured the production function of the cultivated land in the study area based on three indicators: the average land yield of grain, the average land yield of non-food crops, and road network density. The social life function of cultivated land includes providing basic livelihood protection and taking part in leisure and recreation [31], which involves maintaining social stability, necessary livelihood protection, and landscape aesthetics. Thus, in this study, we chose three indicators to measure the living function of the cultivated land in Yuanyang County: the degree of cultivated land contiguity, the distance to the points of interest (POI) of leisure agriculture, and the proportion of agricultural workers. The ecological function refers to the ability of cultivated land to conserve water, purify the environment, control pollution, and maintain the stability of the ecosystem [32]. In this study, four indicators were selected to measure the ecological functions of cultivated land in Yuanyang County: the distance from water bodies, fertilizer and pesticide use per unit area, distance from industrial and mining enterprises, and crop species diversity. Therefore, in this study, we constructed a three-dimensional multifunctional evaluation index system of production, living, and ecology to evaluate the CLUF plots in the study area. In addition, we used the analytic hierarchy process to determine the weight of each index [33] (Table 2).

The graded assignment method was used to assign graded values to the evaluation index of each CLUF (Table 3).

The related indicators were standardized by the range standardization method [34]:

For the positive indicators,
(1)$$r_{ij}=\left(x_{ij}-x_{jmin}\right)/\left(x_{jmax}-x_{jmin}\right).$$

For the negative indicators,
(2)$$r_{ij}=\left(x_{jmax}-x_{ij}\right)/\left(x_{jmax}-x_{jmin}\right).$$

In Equations (1) and (2), $x_{ij}$ is the actual value of the *i*th grid under the *j*th indicator. $r_{ij}$ is the normalized value of the actual indicator. $x_{jmax}$ is the maximum value of the *j*th indicator. $x_{jmin}$ is the minimum value of the *j*th index.

The weighted comprehensive evaluation method was used to calculate the total score of each evaluation unit:(3)$$Q=\overset{n}{\underset{i=1}{\sum }}Q_{i}W_{i}\;$$
where *Q* is the total score of the evaluation unit, *Q_i_* and *W_i_* are the scores and weights of the *i*th indicator, respectively, and *n* is the number of evaluation units.

##### Evaluation of the Leading CLUFs

The leading CLUFs are the result of the interactions between the individual CLUFs, and the NRCA can lift the limitations of time and space to measure the comparative advantages of a research object under different conditions:(4)$$NRCA_{ij}=\frac{E_{ij}}{E}-\frac{E_{i}E_{j}}{EE}$$
where *E_ij_* is the function value of the *j*th function of the *i*th plot, *E* is the sum of the values of all of the function types of all of the plots, *E_i_* is the sum of all of the function values of the *i*th plot, and *E_j_* is the sum of function values of the *j*th function of all of the plots.

### 2.3. Data Sources and Treatment


(1)Statistical data, such as the total output of food crops, the total output of non-food crops, the total sown area of crops, the number of people employed in agriculture, the number of people engaged in agricultural labor, and fertilizer application, were obtained from the Statistical Yearbook of Yuanyang County.(2)Spatial data (Table 4), such as land use data, were obtained from the land use change-based database. Soil organic matter content data, soil profile pattern, groundwater level, and soil pH data were obtained from the Henan Provincial Soil Database. Leaf area index data were obtained from GLOBMAP Leaf Area Index (LAI) Version 3 Description [35]. Leisure agriculture POI data were obtained from the Gaode Map open platform. The Gaode Map application programming interface (API) platform was used to obtain longitude and latitude information of the resort villages, fishing parks, and picking gardens within the study area.


The relevant indicators were treated as follows.

The soil erodibility is mainly related to the soil texture, soil structure, and organic matter content.
(5)$$K=\left[-0.01383+0.51575K_{epic}\right]\times 0.1317$$
(6)$$K_{epic}=\left\{0.2+0.3\exp\left[-0.0256m_{s}\left(1-\frac{m_{silt}}{100}\right)\right]\right\}\times {\left[\frac{m_{silt}}{m_{c}+m_{silt}}\right]}^{0.3}\times \left\{1-\frac{0.25orgC}{\left[orgC+\exp(3.72-2.95orgC)\right]}\right\}\times \left\{1-\frac{0.7\left(1-\frac{m_{s}}{100}\right)}{\left\{\left(1-\frac{m_{s}}{100}\right)+\exp\left[-5.51+22.9\left(1-\frac{m_{s}}{100}\right)\right]\right\}}\right\}.$$

In Equations (5) and (6), *m_c_*, *m_silt_*, *m_s_*, and *orgC* are the percentages of clay particles, powder particles, sand particles, and organic carbon, respectively.

The vegetation cover was calculated using the normalized difference vegetation index (*NDVI*) and the dimidiate pixel method, i.e., a dimidiate pixel model was established using the *NDVI* and the vegetation cover to estimate the vegetation cover of the study area.
(7)$$C=\left(NDVI-NDVI_{min}\right)/\left(NDVI_{max}-NDVI_{min}\right)$$
(8)$$NDVI=\frac{\rho_{2}-\rho_{1}}{\rho_{2}+\rho_{1}}.$$

In Equations (7) and (8), *ρ*_2_ is the pixel value in the infrared band, and *ρ*_1_ is the pixel value in the red band.

Cultivated land contiguity is the degree of concentrated contiguity of the cultivated land plots. The larger the value of cultivated land contiguity, the greater the degree to which the cultivated land is divided, and vice versa.
(9)P=L/A.

In Equation (9), *P* is the degree of cultivated land contiguity, *L* is the total perimeter of the cultivated land in the village area, and *A* is the total area of cultivated land in the village.

Road network density is the density of the roads, and the density of the road network in each grid was calculated by creating a row × column grid of cells.
(10)D=C/T.

In Equation (10), *D* is the road network density, *C* is the length of the roads in the grid cell, and *T* is the area of the grid cell.

The distance from leisure agriculture POI was obtained from the leisure agriculture POI data in the Gaode Map open platform, such as resorts, fishing parks, and picking gardens. Distances of the cultivated land plots from the leisure agriculture POI were obtained using the Near Analysis tool in ArcGIS. The shorter the distance is, the stronger the living function of the cultivated land, and vice versa.

Distance to water bodies: Cultivated land has the ecological function of water conservation, and the distance of the cultivated land plots from water bodies was obtained using the Near Analysis tool in ArcGIS. The shorter the distance is, the stronger the water conservation function.

Distance from industrial and mining enterprises: Cultivated land has the function of pollution control and environmental purification. The distances of the cultivated land plots from industrial and mining enterprises were obtained using the Near Analysis tool in ArcGIS. The shorter the distance is, the stronger the pressure of pollution control, and vice versa.

Crop species diversity is an important basis for farmland ecosystems to remain stable. Shannon’s diversity index was used to estimate crop diversity in this study as follows [36]:(11)$$H^{\prime }=-\overset{s}{\underset{i=1}{\sum }}p^{i}\ln p^{i}$$
where *H′* is the diversity index, *S* is the total number of crops, and *P^i^* is the proportion of the *i*th species to the total area.

## 3. Results

### 3.1. Evaluation of Ecological Security

According to the evaluation results (Figure 3), we used the natural break classification method to classify the ecological security of the study area into three levels: high security level (3.37, 4.43], medium security level (2.81, 3.46], and low security level (0.75, 2.81].

The high security level area in Yuanyang County is 216.17 km^2^, accounting for 30.68% of the total cultivated land area, and is mainly distributed in the central parts of Taiping Town (45.15 km^2^, 20.89%), Luzhai Township (29.20 km^2^, 13.51%), Qijie Town (28.99 km^2^, 13.41%), and the western part of Yuanwu Town (22.99 km^2^, 10.63%). The groundwater level in this area is mostly 1–2 m, the soil organic matter content is high, the soil erodibility is low, and the natural conditions are superior; thus, the ecological security level is high. The medium security level area in Yuanyang County is 305.97 km^2^, accounting for 43.42% of the total cultivated land area, and is mainly distributed in Funingji Town, (44.38 km^2^, 14.5%), Gebukou Township (33.64 km^2^, 11.00%), and the eastern part of Yanga Township (27.35 km^2^, 8.94%). The soil profile pattern of the cultivated land in this area is mainly loam/sand/loam and loam/clay/clay. Loamy soil, which has the advantages of both sandy soil and clayey soil, has excellent cultivation properties and is an ideal soil for crop growth. However, due to the limitation imposed by the soil organic matter content, the soil fertility level is not high; thus, this area has a medium ecological security level of cultivated land. The low security level area in Yuanyang County is 182.53 km^2^, accounting for 25.9% of the total cultivated land area, and is mainly distributed in Funingji Town (23.37 km^2^, 12.81%), Zhulou Township (22.98 km^2^, 12.59%), Guanchang Township (20.34 km^2^, 11.14%), Doumen Township (18.97 km^2^, 10.39%), and Shizhai Town (18.49 km^2^, 10.13%). Compared to the medium or high security level area, the organic matter content in this area is moderately low, and the groundwater level is mostly below 4 m. In addition, due to the low vegetation cover and leaf area index in the area, the buffering effect on the rainfall is poor, and the surface soil is easily eroded by rainfall, while the soil profile pattern is mainly all clay and sand/clay/clay, which is not conducive to rainwater infiltration and plant root extension. Thus, the ecological risk of the cultivated land is high.

### 3.2. Evaluation of CLUFs

#### 3.2.1. Evaluation of Individual CLUFs

The production function of the cultivated land in Yuanyang County is mainly medium to high level, and the overall production function is high. Areas with a high level of production function (Figure 4a) are mainly located in Taiping Town (50.85 km^2^, 16.13%), Shizhai Township (46.57 km^2^, 14.77%), and Zhulou Township (40.66 km^2^, 12.90%), where the grain yield per unit area is higher, the transportation to cultivated land is convenient, and the agricultural mechanization production efficiency is high. Areas with a medium level of production function are mainly concentrated in the areas near the suburban areas such as Funingji Town (67.54 km^2^, 21.30%) and Doumeng Township (39.23 km^2^, 12.37%), where the grain yield is not high due to the profit-seeking effect, but the yield of non-food crops such as vegetables and fruits per unit area is high; thus, the production function of the cultivated land is better. The areas with a low production function are mainly concentrated in Jintang Township (40.66 km^2^, 56.16%) and Dabin Town (24.00 km^2^, 33.15%), where the cultivated land is scattered, which is not conducive to mechanized farming, and the grain yield per unit area is not high; thus, the production function of the cultivated land is low.

The ecological function (Figure 4b) of cultivated land in Yuanyang County is mainly a medium to high level. Areas with a high ecological function are mainly distributed in areas such as Doumen Township (39.74 km^2^, 14.11%) and Dabin Town (34.78 km^2^, 12.35%). These areas are close to the Yellow River; thus, they have a high water-conservation function. However, Funingji Town, which is relatively close to industrial and mining enterprises, faces greater ecological pressure compared to other regional plots and has a strong environmental pollution purification function; thus, the ecological function level in this area is high. Medium ecological function areas are mainly located in the central part of Taiping Town (28.42 km^2^, 10.78%) and the southern part of Shizhai Township (27.86 km^2^, 10.57%), where the use of fertilizers and pesticides per unit area is not high compared to that in other townships, and the amount of external ecological pollution is small; thus, the environmental purification function of the cultivated land is weaker compared to that of the cultivated land in other locations. The lower ecological function areas are mainly located in Qijie Town (41.14 km^2^, 25.77%), the northern part of Luzhai Township (23.72 km^2^, 14.86%), the central part of Gebukou Township (21.98 km^2^, 13.77%), and the eastern part of Funingji Town (16.08 km^2^, 10.07%). Spatially, the most obvious characteristic of areas with a low ecological function level is that they are located in the northern part of Yuanyang County, which is farther away from the Yellow River. Moreover, there are no large rivers or obvious water surfaces inside Yuanyang County; thus, the water conservation function is weak and the ecological function level is not high.

The living function (Figure 4c) of the cultivated land in Yuanyang County is mainly a low to medium level, and the overall living function is not high. Areas with a high living function are mainly concentrated in Yanga Township (43.62 km^2^, 21.13%) and Doumen Township (26.59 km^2^, 12.88%). The cultivated land in this area is closer to leisure agriculture land such as agritainment, resorts, and fishing parks, and the proportion of people engaged in agriculture is large; thus, the living function of the cultivated land is high. The cultivated land with a medium living function is mainly distributed in Taiping Township (42.39 km^2^, 13.78%) and Dabin Town (25.75 km^2^, 8.37%). Although there is leisure agriculture in this area, the proportion of people engaged in agriculture is low, and the living function of the cultivated land is average. Areas with a low living function of cultivated land are mainly located in Funingji Town (52.15 km^2^, 27.35%) and the western part of Shizhai Township (30.71 km^2^, 16.10%), which are located farther away from leisure agriculture facilities and have a low social living function.

#### 3.2.2. Evaluation of Leading CLUFs

NRCA was used to measure the leading CLUFs of each plot. The results (Figure 5) show that the total area of the plots with the leading CLUF of production is 270.51 km^2^, accounting for 38.39% of the total cultivated land area. These areas are mainly concentrated in some parts of Shizhai Township (45.76 km^2^, 16.91%) and Luzhai Township (36.12 km^2^, 13.35%). The cultivated land in these regions mainly has a production function, providing food production, vegetables, and fruit cultivation to ensure food security and improve farmers’ economic benefits. The total area of the plots with the leading CLUF of ecological function is 240.91 km^2^, accounting for 34.16% of the total cultivated land area. These areas are mainly concentrated in the western part of Funingji Town (40.91 km^2^, 16.99%) and Dabin Town (34.61 km^2^, 14.38%). The cultivated land in these areas mainly has an ecological function, with prominent functions of water conservation, pollution control, and environmental purification, ensuring the normal operation of the natural environment and ecosystem. The total area of the plots with the leading CLUF of living function is 193.46 km^2^, accounting for 27.45% of the total cultivated land area. These areas are mainly concentrated in several areas, including Yanga Township (36.57 km^2^, 18.43%) and the western part of Qijie Town (29.92 km^2^, 15.46%). The cultivated land in these areas mainly has the functions of recreation and social security, as well as maintaining social harmony and stability.

### 3.3. Cultivated Land Use Enhancement Zoning

The main function and role of cultivated land is to guarantee food production. Thus, regardless of the ecological security level, the cultivated land with an agricultural production function is classified as the core production zone, the core task of which is to guarantee food production and maintain food security. However, in areas with a high level of ecological security, we classified the cultivated land, for which the social life function is the leading CLUF, as the composite agricultural zone, the task of which is to carry out food production for this area as well as composite agricultural production based on its own advantages in order to increase farmers’ income while ensuring production. In addition, areas where the leading CLUF is the landscape ecological function were classified as the status quo maintenance zone, i.e., the existing production conditions are fully maintained; thus, the ecological quality and cultivated land yield do not decline. Finally, the cultivated land with a low ecological security level and a non-food production function was classified as the natural restoration zone, i.e., focusing on ecological restoration on the basis of agricultural production to stabilize ecological safety (Figure 6).

The spatial overlay analysis of the ecological security level and leading CLUFs was conducted using ArcGIS 10.8 to obtain the zoning of the coordinated development of ecological security and cultivated land use.

### 3.4. Zoning Adjustment and Optimization

The previous zoning results show that some of the townships have scattered and fragmented cultivated land use zoning, which not only increases the difficulty of implementation but also does not facilitate subsequent supervision and management. Therefore, for the convenience of management and the feasibility of implementation, we used the NRCA model again to measure the leading zoning of each township (Table 5, Figure 7) and adjusted the cultivated land use zoning using the nearest neighbor method and the maximum patch principle with the leading zoning of each township as the limiting factor (Figure 8).

The adjusted cultivated land use zoning achieved the effect of spatial clustering, which is not only convenient for implementation and management but can also achieve scale benefits to a certain extent.

Based on the zoning results, the area of the core production zone is 270.52 km^2^, accounting for 38.39% of the total cultivated land area. This zone is mainly concentrated in Shizhai Township (46.69 km^2^, 17.27%), Zhulou Township (40.21 km^2^, 14.87%), Gebukou Township (37.55 km^2^, 13.88%), and Luzhai Township (35.29 km^2^, 13.05%). The core production zone is rich in soil and water resources and has good agricultural production and operation conditions. The cultivated land is concentrated and contiguous, and the yield of agricultural products per unit area is high. The leading CLUF is the production function, and the ecological security level is high. The natural background is superior; thus, it is suitable as the core area of agricultural production for ensuring food production and food security.

The area of the composite agricultural zone is 148.79 km^2^, accounting for 21.21% of the total cultivated land area. This zone is mainly concentrated in Yanga Township (37.90 km^2^, 25.47%) and Qijie Town (31.69 km^2^, 21.29%), in which the cultivated land should be used according to the local conditions to develop composite agriculture and three-dimensional agriculture and to achieve the diversification and enrichment of the agricultural products based on food cultivation. In agricultural production, geographical location and transportation conditions should be taken advantage of to enable the recreational function of the cultivated land to increase farmers’ agricultural income to some extent.

The area of the status quo maintenance zone is 167.56 km^2^, accounting for 23.78% of the total cultivated land area. This zone is mainly located in Dabin Town (26.44 km^2^, 15.78%), the western part of Taiping Town (15.74 km^2^, 26.39%), and the central part of Funingji Town (25.73 km^2^, 15.35%) where the leading CLUF is the landscape ecological function, and the ecological security level is high. It is classified as the status quo maintenance zone for maintaining the ecological security level while carrying out the existing agricultural production and operation activities, as well as continuously maintaining the existing high-quality planting mode, such as growing rice and raising loach together in Taiping Town. This generates more profits than traditional agriculture. It can not only effectively increase the added value of agricultural products, but it can also avoid the use of compound fertilizer, reducing the ecological pressure on cultivated land.

The area of the natural restoration zone is 117.82 km^2^, accounting for 16.72% of the total cultivated land area. This zone is mainly located in Guanchang Township (18.56 km^2^, 15.75%) and Jintang Township (15.00 km^2^, 17.68%). Regarding the spatial location, it is mainly concentrated in the townships along the Yellow River, where the leading CLUF is non-agricultural production, the ecological security level is low, and the ecological risk of the cultivated land is high. Thus, it should be used as an ecological restoration zone. Where conditions permit, rotational cultivation or fallowing should be carried out. Cultivated land planted with two crops a year should be properly planted with one crop to relieve pressure on the cultivated land and promote the self-healing and self-purification of the ecological environment of the cultivated land.

## 4. Discussion

### 4.1. Policy Recommendations

The optimization of cultivated land use cannot be separated from the support of policies. According to the characteristics and problems of different districts, we put forward regional positioning and policy suggestions accordingly.

In the core production zone, the leading CLUF is the production function, but the level of ecological safety is uneven. As demand increases, the use of cultivated land may result in the destruction of the ecological environment in the pursuit of yield. For regions with a low level of ecological security, they should carry out green agriculture and ecological agriculture, and they should promote technologies such as bio-fertilizer and natural enemies for pests to minimize the impact of agricultural production on the environment. For regions with a high level of ecological security, they should improve the effective capacity of cultivated land and not damage the environment during grain production.

In the composite agricultural zone, cultivated land mainly performs living functions and has a high level of ecological security. Diversified food production should be encouraged in composite agricultural areas to ensure grain production while guaranteeing an effective supply of vegetables, fruits, and other types of food, which can also increase the diversity of crop types to a certain extent compared to single food production in order to meet the needs of regional people’s livelihood and the ecological stability of cultivated land.

In the status quo maintenance zone, the leading CLUF is ecological function, and the level of ecological safety is high, which has reached optimal utilization status. Therefore, supervision and guidance should be performed to discriminate the cultivated land use that has a serious negative impact on the ecological environment, and they should allow and promote agricultural production practices with higher environmental sustainability to achieve sustainable use of cultivated land.

In the natural restoration zone, the leading CLUF is ecological function, but the level of ecological safety is low, and it is urgent to restore its own ecological environment quality. Therefore, it is recommended to popularize ecological agriculture awareness and the ecological protection concept and to carry out no-tillage, fallowing, rotational plowing, green manure planting, etc., in order to break the vicious cycle of chemical fertilizer and pesticide residues, cultivated land overdraft, and soil degradation and to maintain the ecological security of cultivated land. At the same time, in order to protect farmers’ income, farmers in ecological restoration areas are given moderate ecological compensation for the costs they pay to protect the cultivated land and to protect their rights and interests.

### 4.2. Advantages and Limitations

Our study proposed the following new perspective: the perspective of foundation-support-threat. This perspective can not only pay attention to the health status of the cultivated land ecosystem itself, but it also takes into account the resilience of its own ecosystem structure and the risks it faces, avoiding the limitation of only selecting statistical indicators, reflecting the spatial form and location of cultivated land, as well as the relationship with adjacent ecosystems, which provided new research ideas for the coordinated development of ecological protection and cultivated land use and also provided a reference for similar regions.

However, there are still some limitations in our research. The first one is the optimization of the evaluation index system. We only paid attention to the material functions of cultivated land on its productive and ecological functions in this study, while for the landscape aesthetics and cultural function of cultivated land [37], only the visual effect of cultivated land’s contiguous degree was used to reflect the aesthetic function of cultivated land, and no good quantitative method has been found for cultural functions. The quantitative index of cultivated land on cultural function will be further explored to make the index system more reasonable in the future. Moreover, our study only explored the coordinated development path of cultivated land use optimization and ecological protection, while the obstacle factors between them were not analyzed. In the future, we will try to analyze the obstacle factors affecting the coordinated development of ecological security and cultivated land use by constructing the obstacle degree model and other methods [38], to further improve and enrich the relevant research content.

## 5. Conclusions

From the perspective of farmland ecological security and leading CLUFs, our study evaluated the level of ecological security and CLUFs in Yuanyang County based on spatial data and statistical data. Then, we constructed cultivated land protection zones based on ecological security levels and leading CLUF zoning. Additionally, we proposed integrated control measures based on local characteristics. The main conclusions are as follows:

First of all, according to the results of the ecological security assessment, Yuanyang County has a high overall level of ecological security. The middle and high level areas are mainly distributed in Taiping Town, Qijie Town, Luzhai Township, Yanga Township, and Yuanwu Town, while the low level of security is mainly distributed in Zhulou Township and Funingji Town. Secondly, from the evaluation results of the leading CLUFs, the cultivated land bearing the production function is mainly distributed in Zhulou Township, Shizhai Town, and Luzhai Township; the cultivated land bearing the ecological function is mainly distributed in Dabin Town, Jintang Township, and the west of Taiping Town; and the cultivated land bearing the living function is mainly distributed in Yanga Township, the western part of Qijie Town, and Jiangzhuang Township. Finally, based on the evaluation results of ecological security and CLUFs, we proposed four cultivated land utilization improvement zones, namely the production core zone, the composite agricultural zone, the status quo maintenance zone, and the ecological restoration zone. Our study can well solve the contradiction between cultivated land use and ecological protection in real life, and, at the same time, developing agriculture according to local conditions can also increase the income of local farmers to a certain extent. Additionally, it can provide new ideas for other countries and regions to achieve ecological protection and the coordinated development of cultivated land use.

## Figures and Tables

**Figure 1 ijerph-19-13630-f001:**
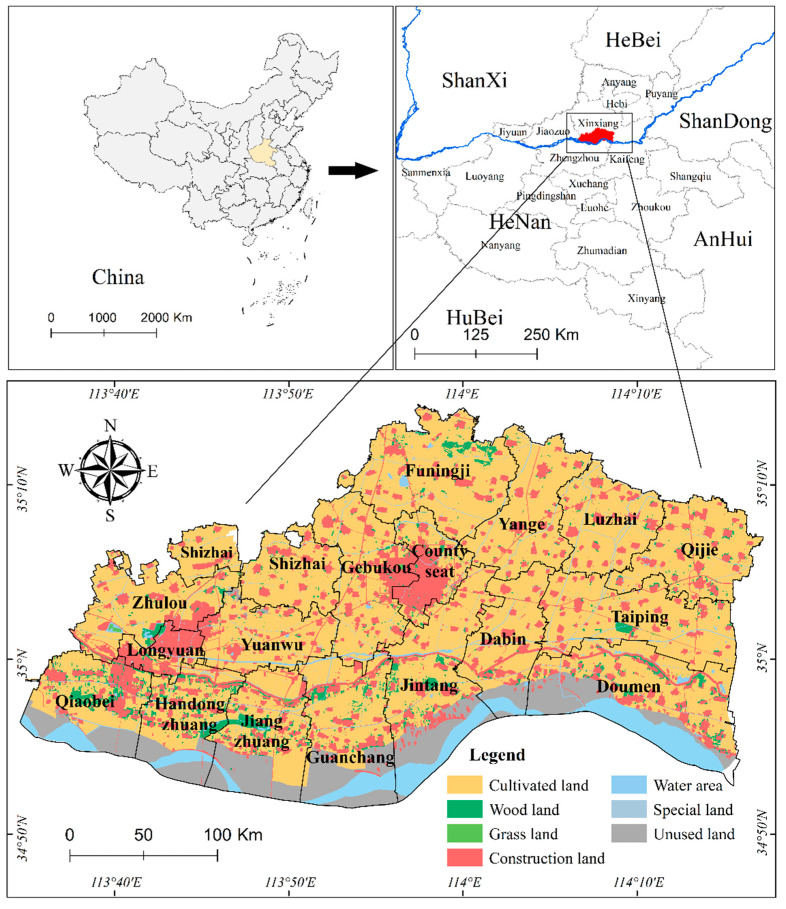
Location of the study area.

**Figure 2 ijerph-19-13630-f002:**
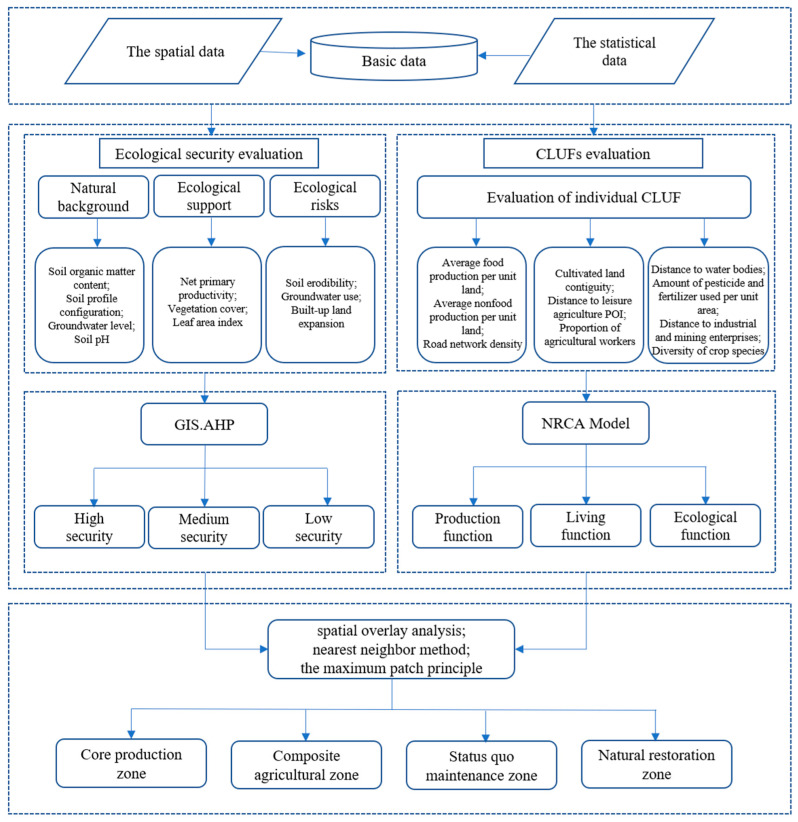
Research framework for optimizing cultivated land use.

**Figure 3 ijerph-19-13630-f003:**
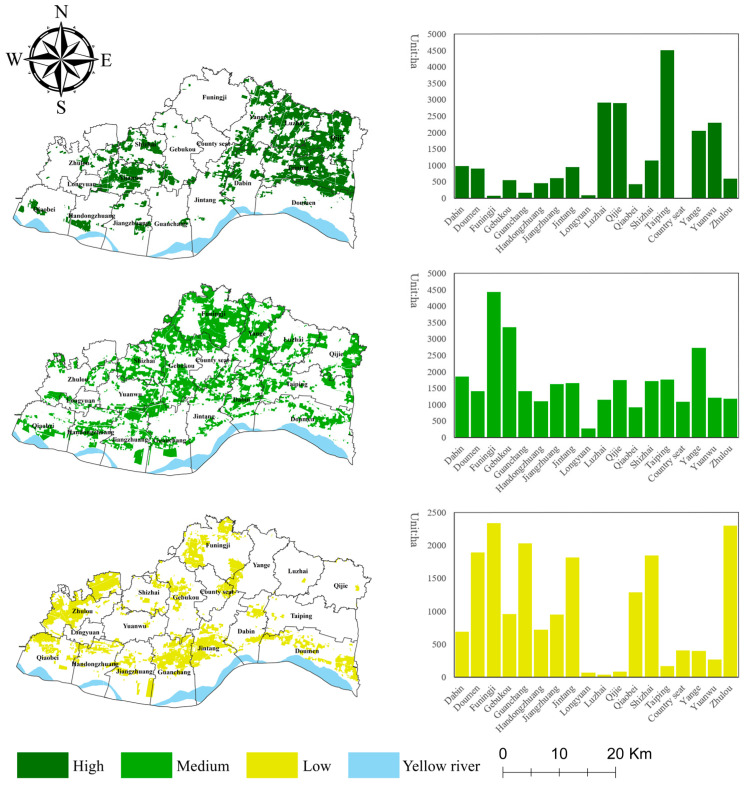
Spatial distribution of the ecological security level of cultivated land in Yuanyang County.

**Figure 4 ijerph-19-13630-f004:**
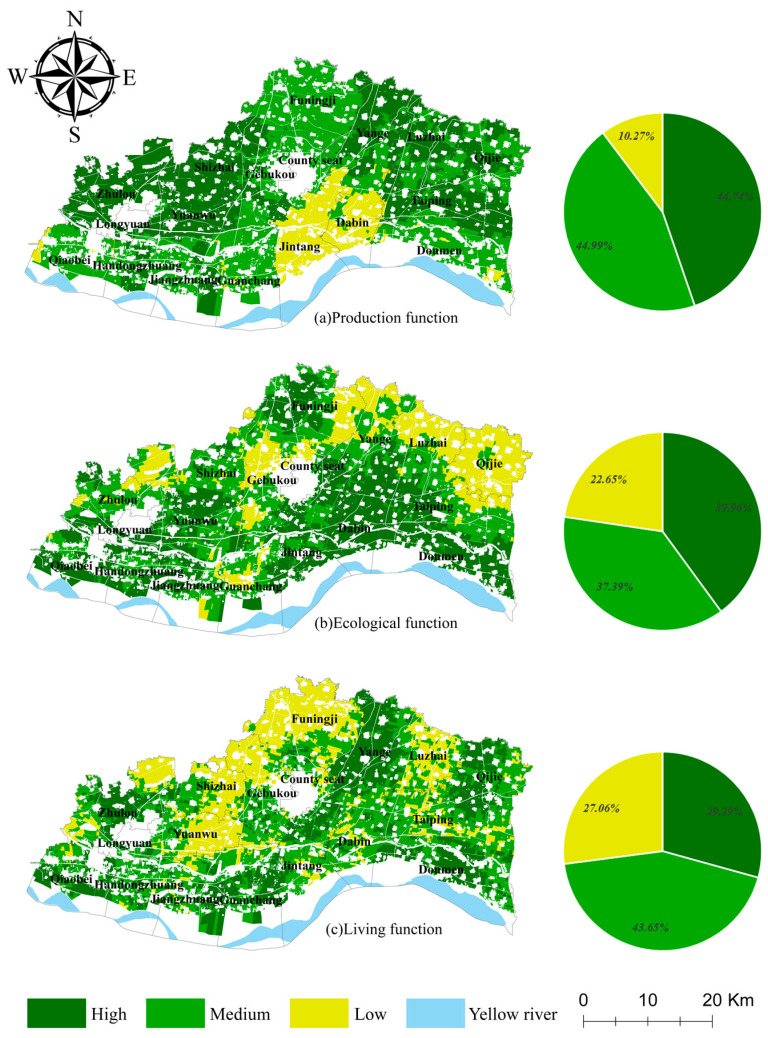
Results of the evaluation of individual CLUFs in Yuanyang County.

**Figure 5 ijerph-19-13630-f005:**
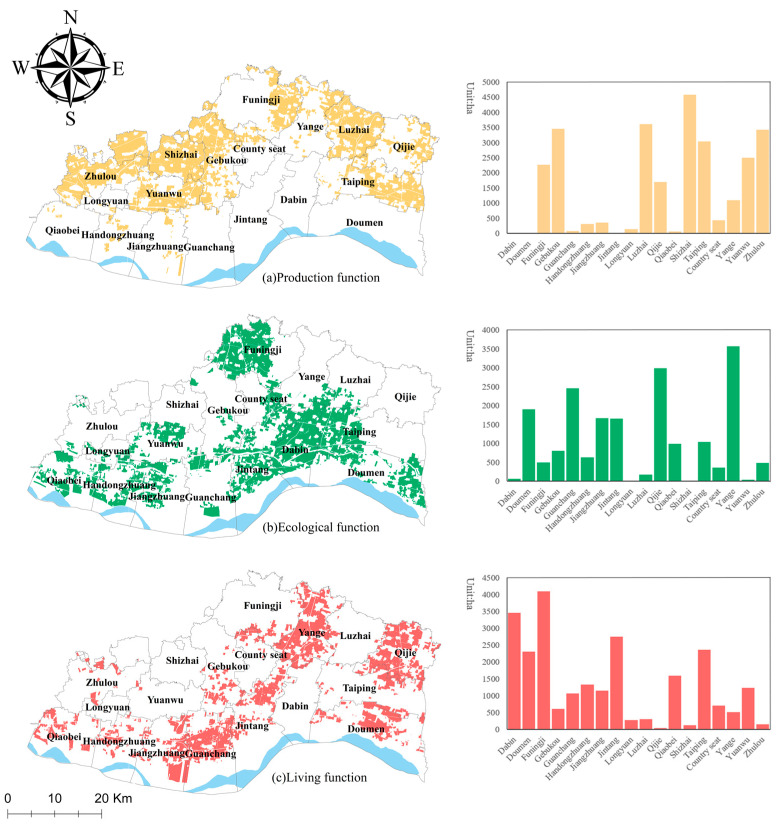
Spatial distribution of the leading CLUFs in Yuanyang County.

**Figure 6 ijerph-19-13630-f006:**
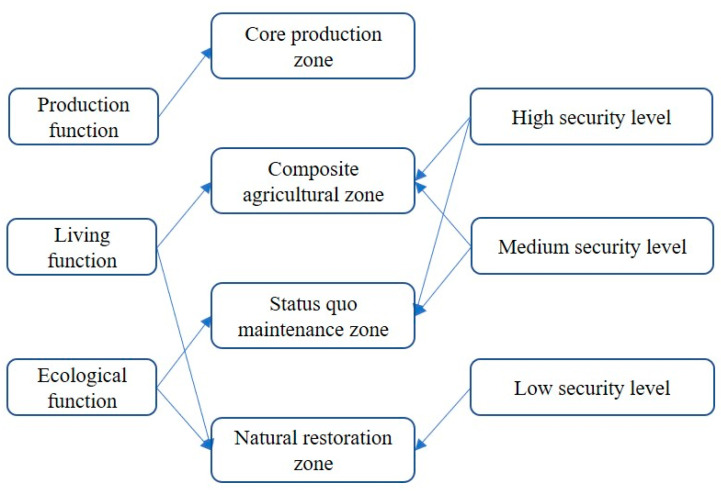
Adjustment of cultivated land use in Yuanyang County.

**Figure 7 ijerph-19-13630-f007:**
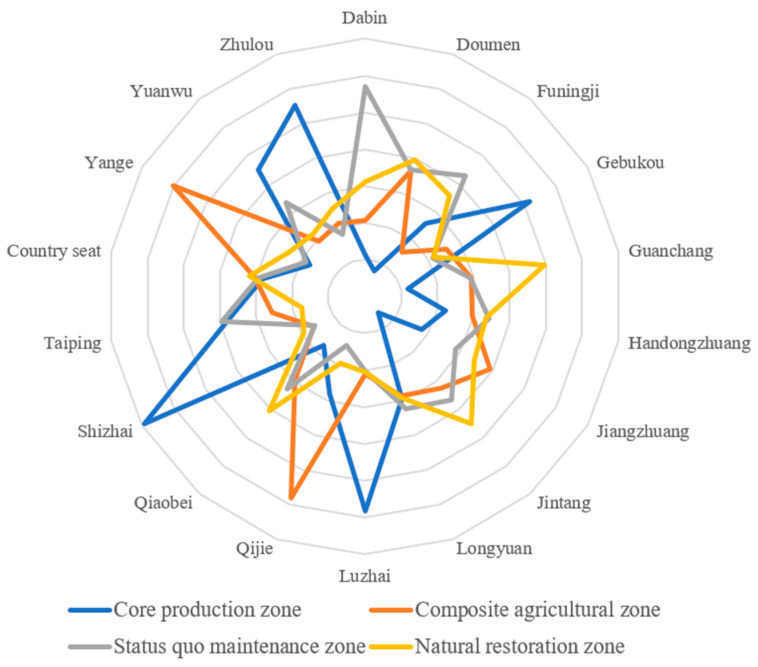
NRCA value of villages and towns in Yuanyang County.

**Figure 8 ijerph-19-13630-f008:**
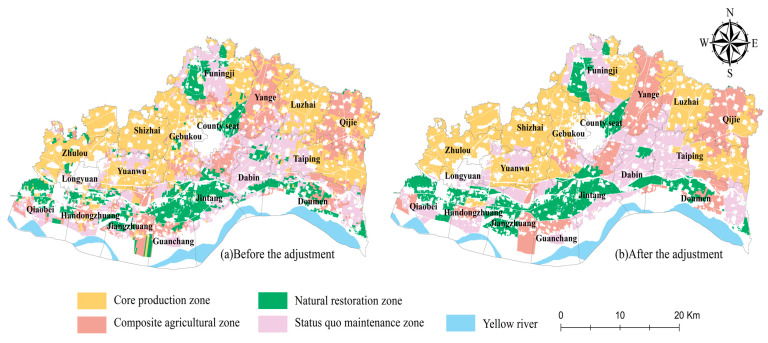
Zoning map for the adjustment of cultivated land use in Yuanyang County.

**Table 1 ijerph-19-13630-t001:** Yuanyang County cultivated land ecological security evaluation index system.

Primary Indicator	Secondary Indicator	Score				
		1	2	3	4	5
Natural background	Soil organic matter content (%)	(0, 0.6]	(0.6, 1.2]	(1.2, 1.8]	(1.8, 2.4]	(2.4, +∞)
	Soil profile configuration	All clay	Sand/clay/clay	Soil/sand/soil	Soil/clay/clay	All soil or soil/clay/soil
	Groundwater level (m)	[4, +∞)	(2, 4)	1	(1, 2)	2
	Soil pH	(8.5, 9.5)	-	[6.5, 7.5]	-	(7.5, 8.5]
Ecological support	Net primary productivity (kg·C/m²)	(0, 0.1]	(0.1, 0.2]	(0.2, 0.3]	(0.3, 0.4]	(0.4, 1]
	Vegetation cover	(0, 0.2]	(0.2, 0.4]	(0.4, 0.6]	(0.6, 0.8]	(0.8, 1]
	Leaf area index	(0, 0.2]	(0.2, 0.4]	(0.4, 0.6]	(0.6, 0.8]	(0.8, 1]
Ecological risks	Soil erodibility	(0.8, 1]	(0.6, 0.8]	(0.4, 0.6]	(0.2, 0.4]	(0, 0.2]
	Groundwater use	-	Groundwater overdraft zone	-	Undrawn groundwater zone	-
	Built-up land expansion (km)	(0, 0.5]	(0.5, 1]	(1, 2]	(2, 4]	(4, +∞)

**Table 2 ijerph-19-13630-t002:** Evaluation index system and weights of CLUFs in Yuanyang County.

Type	Indicator	Unit	Weight	Meaning	Positive(+)/Negative(−)
Production function	Average food production per unit of land	t/hm^2^	0.43	Food supply capacity of cultivated land	+
	Average non-food production per unit of land	t/hm^2^	0.32	Vegetable and fruit supply capacity of cultivated land	+
	Road network density	km/km^2^	0.25	Plot accessibility	+
Living function	Cultivated land contiguity	-	0.35	Landscape aesthetics	−
	Distance to leisure agriculture POI	km	0.28	Recreational potential of cultivated land	−
	Proportion of agricultural workers	-	0.37	Social security capacity of cultivated land	+
Ecological function	Distance to water bodies	km	0.16	Water conservation capacity	−
	Amount of pesticide and fertilizer used per unit area	t/hm^2^	0.27	Environmental purification capacity	+
	Distance to industrial and mining enterprises	km	0.21	Pollution control capacity	−
	Diversity of crop species	-	0.36	Farmland ecosystem stability	+

**Table 3 ijerph-19-13630-t003:** Classification and assignment of CLUF evaluation indicators in Yuanyang County.

Type	Indicator	Score				
		1	2	3	4	5
Production function	Average food production per unit of land	(0, 6.57]	(6.57, 6.85]	(6.85, 7.26]	(7.26, 7.44]	(7.44, +∞)
	Average non-food production per unit of land	(0, 10.63]	(10.63, 14.75]	(14.75, 21.93]	(21.93, 37.93]	(37.93, +∞)
	Road network density	(0, 2.71]	(2.71, 5.27]	(5.27, 10.11]	(10.11, 40.25]	(40.25, +∞)
Living function	Cultivated land contiguity	[0.046, 1)	[0.026, 0.046)	[0.02, 0.026)	[0.017, 0.02)	(0, 0.017)
	Distance to leisure agriculture POI	(5.16, +∞)	(3.74, 5.16]	(2.53, 3.74]	(1.4, 2.53]	(0, 1.4]
	The proportion of agricultural workers	(0, 0.2]	(0.2, 0.4]	(0.4, 0.6]	(0.6, 0.8]	(0.8, 1)
Ecological function	Distance to water bodies	(2.95, +∞)	(1.85, 2.95]	(1.7, 1.85]	(0.51, 1.07]	(0, 0.51]
	Amount of pesticide and fertilizer used per unit area	(0, 0.67]	(0.67, 1.02]	(1.02, 1.62]	(1.62, 2.08]	(2.08, +∞)
	Distance to industrial and mining enterprises	(5.96, +∞)	(4.12, 5.96]	(2.61, 4.12]	(1.38, 2.61]	(0, 1.38]
	Diversity of crop species	(0, 0.2]	(0.2, 0.4]	(0.4, 0.6]	(0.6, 0.8]	(0.8, 1)

**Table 4 ijerph-19-13630-t004:** Data description used for ecological security evaluation and CLUF evaluation.

Data Names	Data Sources
Land use maps	Land use change-based database
Soil organic matter content	Henan Provincial Soil Database
Soil profile configuration	Henan Provincial Soil Database
Groundwater level	Henan Provincial Soil Database
Soil components	Chinese soil dataset based on the Harmonized World Soil Database (http://www.iiasa.ac.at/Research/LUC/External-World-soil-database/HTML/ accessed on 2 September 2022)
Soil pH	Henan Provincial Soil Database
Net primary productivity (NPP)	MOD17A3HGF Version 6.0 (https://lpdaac.usgs.gov accessed on 2 September 2022)
Normalized difference vegetation index (NDVI)	Resource and Environment Science and Data Center (http://www.resdc.cn accessed on 2 September 2022)
Leaf area index	GLOBMAP Leaf Area Index (LAI) Version 3 Description (http://www.resdc.cn accessed on 2 September 2022)
Leisure agriculture POI	Gaode Map open platform
Road network	Land use change-based database
Statistical data	Statistical Yearbook of Yuanyang County

**Table 5 ijerph-19-13630-t005:** Calculation results of the NRCA value of each township in Yuanyang County.

	Core production Zone	Composite Agricultural Zone	Status Quo Maintenance Zone	Natural Restoration Zone
Dabin	−0.019	−0.009	0.027	0.001
Doumen	−0.022	0.006	0.007	0.009
Funingji	−0.004	−0.014	0.013	0.006
Gebukou	0.022	−0.004	−0.009	−0.009
Guanchang	−0.018	−0.001	−0.001	0.020
Handongzhuang	−0.008	0.000	0.004	0.003
Jiangzhuang	−0.012	0.009	−0.001	0.004
Jintang	−0.024	0.002	0.007	0.015
Longyuan	0.000	−0.001	0.002	−0.001
Luzhai	0.028	−0.009	−0.010	−0.009
Qijie	−0.002	0.028	−0.016	−0.011
Qiaobei	−0.013	−0.001	0.003	0.010
Shizhai	0.039	−0.014	−0.014	−0.011
Taiping	0.008	−0.005	0.009	−0.013
Country seat	−0.002	0.000	−0.001	0.002
Yange	−0.013	0.030	−0.011	−0.006
Yuanwu	0.015	−0.010	0.003	−0.008
Zhulou	0.025	−0.009	−0.012	−0.004

## Data Availability

Not applicable.

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
