# Peer review of "Optimizing the Use of Cultivated Land in China’s Main Grain-Producing Areas from the Dual Perspective of Ecological Security and Leading-Function Zoning"

_ijerph, 2022, doi:10.3390/ijerph192013630_

Round 1
Reviewer 1 Report
The manuscript discussed a very interesting topic on the optimization of cultivated land use in China's Main Grain producing Areas from the perspective of ecological security and leading-function zoning. Overall, this is a well-developed piece of work including conceptualization, data analysis and writing. I have few suggestions for further improvements.
1. In the introduction part, the author briefly discussed the ecological security, cultivated land functions. However, the research progress on the optimization of cultivated land use can be strengthened in this part to make the introduction smoother.
2. The author should stress the contribution of this study further comparing with existing studies in Discussion section, which can highlight the scientific value of this study and make it easier for the reader to understand.
3. How to optimize arable land use from the perspective of ecological security and leading-function zoning is an important issue addressed in this paper. However, the explanation of its research method is still imperfect at present, so it is suggested to introduce the research ideas and technical outline of the paper in the research methodology section.
4. Conclusions should be more concise and condensed. The first two paragraphs are equivalent to enumerating the results again. Some of the content is even similar to the content in Abstract. And I suggest that if the author summarize the laws and enlightenment behind it based on the research results.
5. The whole manuscript is recommended to be polished with more authentic language. For instance, there is a confusion between LUFs and LUFs in line 278 “2.2.2. LUF evaluation “and line 279 “(1) Evaluation of individual LUFs” should be LUFs evaluation, and individual LUF.
Would it be more accurate for “cultivated land use functions (LUFs)” to modify it to “cultivated land-use functions (CLUFs)”?
Reviewer 2 Report
1.Why choose Yuanyang County as the study area?I can not find its value from the existing article.In the part of “Overview of the study area”,The author simply describes the geographic location of Yuanyang County
2.The paper has many small errors.1)Lin 210:this county has a 210 beach area of 33,666.67 hm2, including 22,533.33 hm2,the spelling of the “hm2” is right? 2)There is a problem with the scale of the Chinese map,the author shoule visit this website: https://www.tianditu.gov.cn/.
3.In the part of discussion, “most previous 556 studies have used the PSR model to evaluate the ecological security level of the cultivated 557 land by selecting several natural, economic, and environmental statistical indicators re- 558 lated to the level of input and output, the infrastructure conditions, and the utilization of 559 the cultivated land. It is true that this method can better reflect the coordination relation- 560 ship between the economy, society, and environment……” Why do you still describe previous research progress in the part of discussion?The author should learn how to write the part of discussion.
4. The author propose the following recommendations.In fact, the author's suggestions can be placed in other county.
Reviewer 3 Report
Comments and Suggestions for Authors
The ecological security and cultivated land use functions (LUFs) in the study area are evaluated by constructing a comprehensive evaluation index system. In general, the research has its own innovative points and the system is complete, but there are still obvious problems. The discussion of the results is not in-depth in this paper, the citation of references is up to 2.2.2, and none of the references is cited in the discussion, which is far from sufficient for a research article. It is suggested to reorganize the article discussion to better reflect the necessity of research. In addition, the conclusion part is highly similar to the result part, and it is recommended to reorganize.
The following is a list of problems, please refer to the annotations in the pdf file for details.
Detailed comments
1. Introduction
The part of the preface is too long, it is recommended to reorganize it, and pay attention to the logical relationship between the various paragraphs.
2. Materials and Methods
The article does not clarify the basis for the establishment of the weights in Table 2.
In Table 2, there are positive indicators and negative indicators in the evaluation of ecological function, which are not clarified in the table, and the processing methods of the two indicators are not mentioned in the calculation process.
3. Results
Line 386-406
From the picture, we can see that the area of low safety level in Funingji is also relatively large. Please confirm whether there is any mistake in this conclusion.
In addition, it is suggested to add brackets after the main distribution villages, and supplement the information such as area or proportion, to make up for the shortcoming that the area distribution size of some towns and villages cannot be clearly defined in the picture.
Line 400
This description is a bit far-fetched. Some of the middle and safe areas are also distributed along the Yellow River. Please re-judge the main distribution towns of each area based on figure 2.
4. Discussion
Some of the points discussed are not derived from the results of this paper, but no references are cited. Suggestions are added to make the discussion part more convincing.
The discussion part is not in-depth enough, just around the results of the phenomenon of the cause analysis, did not continue to dig the results of this study and other research results similar or differences point out, advantages or disadvantages. It is partly similar to the research background.
Line 633-635
If this conclusion is based on the work of other scholars, please add references.
5.Other detailed comments
Please check that the writing of hm2 in the full text is correct.
Line 297
A total of 65 references are listed in this paper. Please check.
Please check that the references cited in the full text are accurate.

Round 2
Reviewer 2 Report
1.The author should learn how to write the part of discussion.
2.It is recommended that the author change the study area.
3.Moderate English changes required.
Reviewer 3 Report
The paper as a whole has been greatly changed, and there are still some questions and details that need to be carefully considered.
1. Most indicators are used in the section "2.2.2 Ecological security evaluation ". The author is requested to supplement the data sources of indexes including soil physical and chemical properties indexes, and clarify how to obtain these original data, so as to make the evaluation results of the paper more credible.
2. Please pay attention to the details of the article. For example, if a word in line 530 is misspelled, check the full text carefully.
Round 3
Reviewer 2 Report
As we all know, IJERPH is a international journal.From a global perspective, it is recommended that the author should change the study area, the research results also have not strong reference significance for similar regions
